# Comparative Fungicidal Activities of N-Chlorotaurine and Conventional Antiseptics against *Candida* spp. Isolated from Vulvovaginal Candidiasis

**DOI:** 10.3390/jof8070682

**Published:** 2022-06-28

**Authors:** Mayram Hacioglu, Ozlem Oyardi, Fatima Nur Yilmaz, Markus Nagl

**Affiliations:** 1Department of Pharmaceutical Microbiology, Faculty of Pharmacy, Istanbul University, Istanbul 34116, Turkey; f.yilmaz@istanbul.edu.tr; 2Department of Pharmaceutical Microbiology, Faculty of Pharmacy, Gazi University, Ankara 06330, Turkey; ozlemoyardi@gazi.edu.tr; 3Institute of Hygiene and Medical Microbiology, Medical University of Innsbruck, 6020 Innsbruck, Austria

**Keywords:** vulvovaginal candidiasis, *Candida* spp., antiseptic, N-chlorotaurine, fungicidal activity

## Abstract

N-chlorotaurine (NCT), the N-chloro derivative of the amino acid taurine, is a long-lived oxidant produced by stimulated human leucocytes. NCT has antimicrobial activities which are generally enhanced in the presence of organic material. The aim of this study was to investigate fungicidal effects of NCT and conventional antiseptics against *Candida* isolated from vulvovaginal candidiasis (VVC). Chlorhexidine (CHX, 1.6%), octenidine dihydrochloride (OCT, 0.08%), povidone iodine (PVP-I, 8%), boric acid (8%), and NCT (0.1% (5.5 mM)) were evaluated against forty-four *Candida* isolates, according to European Standard methods, at 30, 60, 90, and 120 min and 24 h in the presence of skim milk as an organic material. CHX, OCT, and PVP-I showed rapid fungicidal activity against all *Candida* isolates with 5–6 log_10_ reduction of viable counts after 30 min, whereas boric acid and NCT needed 1 h against *Candida albicans* and 2 h against *non-albicans Candida* for a significant 3 log_10_ reduction. NCT showed fungicidal activity (defined as ≥4 log_10_ reduction) against *C. albicans* within 90 min and *C. non–albicans* within 24 h. Based upon all presently available data, including our results, NCT could be used as a new agent for treatment of local fungal infections such as VVC.

## 1. Introduction

The vaginal microbiota is a dynamic healthy ecosystem and this ecosystem is defined by an acidic environment (pH less than 4.5) which is unsuitable for most bacteria, fungi, and viruses. Loss of the acidic environment may cause vaginal infections. Some vaginal infections are symptomatic and some of the most common symptoms may be discharge, itching, or odor. The most common vaginal infections are vulvovaginal candidiasis (VVC), bacterial vaginosis, and trichomoniasis, which is a sexually transmitted infection [1,2]. VVC usually is caused by *Candida albicans* but can also be caused by *non–albicans Candida* species, especially *Candida glabrata* [3].

More than half of all sexually mature women suffer at least one episode of VVC, and 40–45% of them have two or more episodes [4]. Based on clinical presentation, microbiology, host factors, and response to treatment, VVC can be classified as uncomplicated or complicated. Three or more symptomatic episodes of VVC in <1 year affecting <5% of women are defined as recurrent VVC that may be either idiopathic or secondary (related to frequent antibiotic use, diabetes, or other underlying host factors) [2].

Despite considerable patient morbidity, options for the treatment of VVC and recurrent VVC (particularly in the setting of azole resistance) are limited. Antifungal agents used for the treatment of VVC are orally and topically available imidazoles (e.g., clotrimazole, butoconazole, miconazole), triazoles (e.g., fluconazole, itraconazole), and polyenes (e.g., nystatin and amphotericin B) [2,3]. Azole drugs, such as fluconazole, have clinically been widely used for treatment of VVC and recurrent VVC, and although fluconazole is a primary therapeutic option for the treatment of these infections, increased use and misuse has resulted in azole resistance among *Candida* spp. isolated from vulvovaginitis patients. Recent studies showed that fluconazole resistance among these isolates may correlate with CDR1 gene overexpression [5,6].

Alternative treatments for vaginal infections include antiseptic agents, such as boric acid, chlorhexidine (CHX), octenidine dihydrochloride (OCT), and povidone iodine (PVP-I), each with different mechanisms of action. They are safe if administered at appropriate concentrations. Most of the vaginal antiseptics are available in the form of a vaginal douche [1]. These agents have some advantages compared to antifungal drugs, such as (1) they have faster fungicidal activity, (2) their resistance development is much less frequent or absent because of their multitarget mechanism of action, (3) they can also be used for mixed vaginal infections due to their broad antimicrobial spectrum, and (4) they can be used for pre- and postoperative prophylaxis [7].

Boric acid is an inorganic acid which is known for its antibacterial, antifungal, and antiviral activity and its antiseptic and astringent characteristics, and it has been used for decades to treat vulvovaginal and ear infections. It has become a first-line alternative to azoles in the context of resistance, and 600 mg of boric acid administered vaginally once daily in a gelatin capsule for three weeks is currently recommended by the US Centers for Disease Control and Prevention (CDC) for treatment of recurrent *C. non-albicans* VVC [2,8,9]. Using it as a topical powder can potentially help to control fungal growth, relieve itching and inflammation, and speed up the healing process [8]. 

CHX is a bisbiguanide antiseptic and preservative with a broad spectrum of antibacterial and antifungal activity used especially in the prevention and treatment of oral mucosal infections. It was found that 0.25–0.5% of CHX vaginal douches demonstrated efficacy against bacterial and fungal vaginitis [10,11]. 

Another alternative for the treatment of resistant cases is PVP-I. It is a broad-spectrum antiseptic and has germicidal effects against Gram-positive and Gram-negative bacteria, viruses, and fungi. It has been in use for over 60 years as a topical solution, ointment, or vaginal suppository. Clinical applications of PVP-I include antisepsis of the skin, wounds, oral cavity, eyes, intrasurgical lavage, and vagina [8,12]. 

OCT is a cationic active compound that is well established as skin, mucous membrane, and wound antiseptic solution, and, additionally, it is recommended as an alternative in case of triclosan resistance. Moreover, OCT has broad-spectrum activity, including common pathogens of wound infections and multidrug resistant bacteria [13]. Today, it is an established antiseptic in a wide range of applications and represents an alternative to older substances such as CHX, PVP-I, or triclosan [14]. It was recently shown to be effective even at lower doses than commercially available concentrations against the multidrug-resistant yeast *Candida auris*, which causes significant outbreaks in hospitals and infections with high mortality [15]. 

N-chlorotaurine (Cl–HN–CH_2_–CH_2_–SO_3_-, NCT), the N-chloro derivative of the amino acid taurine, is a long-lived oxidant produced by activated human granulocytes and monocytes. Besides immune modulatory effects, NCT has bactericidal (Gram-positive and Gram-negative bacteria), virucidal (herpes simplex, adenoviruses, influenza, SARS-CoV-2 (COVID-19), respiratory syncytial virus, human immune deficiency virus), protozoocidal (amoebae, leishmaniae, and trichomonads), and also fungicidal (yeasts and molds) activities [16,17]. NCT is a mild, long-lived oxidant that can be applied to sensitive body regions as an endogenous antiseptic. The pure crystalline sodium salt of NCT (MW = 181.57 g/mol) can be chemically synthesized [18]. Due to its unspecific oxidative mechanism of action, the development of resistance is extremely unlikely and actually has not been detected in laboratory tests [16,17,18].

Although the antifungal properties of NCT are known, studies investigating its efficacy against *Candida spp*. are few [19,20]. Therefore, the purpose of this study was to determine the in vitro activities of NCT and conventional antiseptics against *Candida* isolates from patients with VVC and to compare them.

## 2. Materials and Methods

### 2.1. Test Microorganisms

Forty-four strains of *Candida* isolates were used for the experiment. All strains were collected from patients diagnosed with VVC at the Clinical Microbiology Laboratories of the Group Florence Nightingale Hospitals in Turkey. Twenty-nine of the strains were *C. albicans*, whereas fifteen of them were *C*. *non-albicans*. The total number of strains is shown in Table 1 (fluconazole resistance profiles were conducted in a previous study [21]).The yeasts were identified by VITEK 2 (BioMerieux, Craponne, France) and CHROMagar, and verified by API 20C AUX (BioMerieux, France) systems. Prior to analysis, each isolate was cultured on Sabouraud dextrose agar (SDA, Merck) plates to ensure viability. *C. albicans* ATCC 10231 was used as the quality control strain.

### 2.2. Test Products

Five different antiseptics were included in the study. NCT was manufactured at the Institute of Hygiene and Medical Microbiology at the Medical University of Innsbruck, Austria [18]. NCT as a crystalline sodium salt (molecular weight 181.57 g/L) was prepared to be pharmaceutical quality using chloramine T and taurine at the Institute of Hygiene and Medical Microbiology at the Medical University of Innsbruck, Austria [18]. The used lot from 27 November 2017 was sterile and pyrogen-free with a potency of >99%. Final concentration of NCT used in experiments was 0.1% (1 mg/mL, 5.5 mM). Standard concentrations of antiseptics that were within the range of use or possible use were chosen. Commercially available products CHX (2%, Microvem, Altun Sterilizasyon ve Medikal), OCT (0.1%, Octenisept, Schülke & Mayr GmbH), and PVP-I (10%, Batiqon, Alfa Temizlik Medikal ve Sağlik. Ürünleri İmalatı) were provided from manufacturers. Boric acid was dissolved from dry powder with distillated water (10%, Merck).

### 2.3. Antiseptic Activity Assay

Quantitative killing tests used to assess fungicidal activity were performed as described in the European Standards with some modifications [22,23]. Yeasts were grown in SDA for 24 h. After incubation, cultures were collected into sterile physiological buffered saline (PBS), centrifuged (about 5000× *g*, 5–10 min), and washed with PBS. Cells were suspended in PBS until a cellular density equivalent to 1–3 × 10^8^ cfu/mL of the McFarland standard was reached. The dilutions of the yeasts were prepared in PBS and spread on duplicate SDA and plates were incubated at 37 °C for 48 h. At the end of the incubation, emergent colonies were counted and the number of cfu/mL were determined to confirm the yeast inoculum. 

Initially, suspensions of 1–3 × 10^8^ cfu/mL of yeast were prepared, but after the dilutions (adding skim milk and the test antiseptics, see below) 1–3 × 10^7^ cfu/mL was the final fungal concentration in the samples. An amount of 1 mL of yeast inoculum was added to 1 mL of 10% skim milk (Merck) as organic substances and incubated for 2 min at room temperature. Then, 8 mL of antiseptic solution was added to the mixture and incubated for 30, 60, 90 and 120 min and 24 h at 37 °C and pH 7. Final concentrations of antiseptics were 0.1% NCT, 1.6% CHX, 0.08% OCT, 8% PVP-I, and 8% boric acid. After incubation, 1 mL of suspension was transferred into 8 mL of neutralizer solution and 1 mL of distillated water. The neutralizer solution contained lecithin 3 g/L, polysorbate-80 30 g/L, sodium thiosulfate 5 g/L, and L-histidine 1 g/L, and the pH was adjusted to 7.0. After 5 min of neutralization process, 1 mL of the solution was transferred to plates in duplicate and melted SDA was poured onto the plate. Plates were incubated for 48 h at 30 °C to count yeast colonies and calculate the number of living yeasts. A reduction in cfu/mL by least 4 log_10_ was determined as fungicidal activity. 

### 2.4. Validation Procedures

The efficacy of inactivation of test agents and absence of antimicrobial effects of the neutralizer were evaluated according to the European Standard Methods to validate the test results [23].

### 2.5. Statistics

The data are presented as mean values and standard deviations (SD) or standard error of the mean (SEM) of independent experiments with 29 strains of *C. albicans* and 15 strains of *C. non-albicans*. The Kruskal–Wallis test was used to test for a difference between the test and control groups. A *p*-value of <0.05 was considered significant for all tests. Calculations were performed with the GraphPad Prism 8.0.1 software (GraphPad, Inc., La Jolla, CA, USA).

To gain an improved survey on the microbicidal activity of the different antiseptics against *C. albicans* and *C. non-albicans*, the recently introduced integral method was used, which transforms the whole killing curve (log_10_ CFU/mL versus time) into one value of “Bactericidal Activity (BA, log_10_ CFU ml^−1^ min^−1^)” [24]. The method allows an expanded statistical analysis, particularly between killing curves with small differences. One-way analysis of variance (ANOVA) and Tukey’s multiple comparison test were used to test for a difference between BA values of the antiseptics.

## 3. Results

A total of forty-four strains of Candida isolates were collected from patients with VVC. Among these strains, 29 were identified as *C. albicans*, whereas there were 8 *C. glabrata*, 2 *Candida tropicalis*, 2 *Candida kefyr*, 1 *Candida krusei*, 1 *Candida famata*, and 1 *Candida lusitaniae* that were identified (Table 1).

The quantitative killing assays with five antiseptics against *Candida* isolates were determined according to the European Standards. The percentages of yeast strains reduced by ≥4 log_10_ after the respective incubation time are summarized in Figure 1 and Figure 2. According to these results, 0.08% OCT and 1.6% CHX were the most effective agents against *C. albicans* within 30 min. PVP-I (8%) and 0.1% NCT killed all *C. albicans* isolates at 60 and 90 min, respectively, by ≥4 log_10_. Among the antiseptics, only 8% boric acid did not lead to a ≥4 log_10_ reduction against all isolates after 24 h (Figure 1), but had a little lower activity against one isolate each of *C. albicans* and *C. non-albicans* with a log_10_ reduction of 3.60 and 3.52, respectively.

OCT, CHX, and PVP-I were more effective antiseptic agents against *C. glabrata* and other *C. non–albicans* within 30 min than NCT and boric acid, which needed 24 h to kill the yeasts to the detection limit. (Figure 2).

The killing curves are shown in Figure 3 and Figure 4. The rapid decrease in viable fungal counts by OCT, CHX, and PVP-I is clearly visible with no statistical difference between these three antiseptics and with no difference between *C. albicans* and *C. non-albicans*. The killing was significantly slower with boric acid and NCT (*p* < 0.01), which showed similar curves (*p* > 0.05). The activity of boric acid and NCT was slightly higher against *C. albicans* than against *C. non-albicans*. The killing curve of NCT against different strains of *C. albicans* was much more uniform than that of boric acid. In *C. non-albicans*, great differences between single strains occurred for both of these antiseptics. Except *C. tropicalis* strains, NCT showed quicker activity than boric acid against non-albicans strains. Against eight *C. glabrata* strains, both disinfectants showed similar effects overall, but different effects for the individual strains.

Validation of inactivation activity and absence of antifungal activity of the neutralizer was within the margin of error, according to the European Standards.

The killing curves were analyzed with the integral method (Table 2). Because of the large number of isolates, the small differences in the course of the killing curves expressed as BA values also turned out to be significant between all test antiseptics at their individual test concentrations used.

## 4. Discussion

It has been estimated that 50–75% of women will experience VVC at some time in their lives [25]. Antiseptics are currently available on the market, which can be used for adjunctive treatment of VVC, such as boric acid, CHX, OCT, and PVP-I. In the present work, five different antiseptics were investigated against forty-four strains of *Candida* isolates which were collected from patients with VVC. CHX, OCT, and PVP-I showed rapid fungicidal activity against all *Candida* isolates within 30–60 min, whereas boric acid and low-concentrated NCT needed a few hours for a reduction of viable counts by ≥4 log_10_ steps. 

CHX is a bisbiguanide antiseptic with a broad spectrum of antimicrobial activity and many in vitro and in vivo studies have shown that CHX is very effective against *Candida* infections, including vaginal infections [10]. Additionally, it has been shown that the use of 2% CHX gluconate as a vaginal preoperative preparation was safe and was not associated with increased vaginal irritation or allergic reactions [26]. Previous studies have shown that CHX is effective against vaginal infections of either bacterial or fungal origin, both in vitro and in vivo mouse and rat vaginitis infection models [10,11].

OCT and PVP-I have broad spectrum activity and are used as skin antiseptics against common pathogens. Many researchers have shown that OCT and PVP-I have antimicrobial activity against various yeasts including *Candida*, which are isolated from VVC [27,28,29]. Consistent with these results, we found that CHX, OCT, and PVP-I were fungicidal against a variety of *Candida* isolates in the present work. 

It is known that boric acid is a safe, alternative, economic option for women with recurrent and chronic symptoms of VVC even when conventional treatment fails [30]. De Seta et al. [31] showed that 5% boric acid was fungicidal against *C. albicans* strains, and it also interfered with the development of biofilm and hyphal transformation after 24 h of incubation. Our results also showed that it was effective against all *Candida* strains at 8% concentration within 24 h.

NCT has killing activity against bacteria, fungi, viruses, and parasites, and the 1% aqueous solution can be used as an antiseptic and applied to the eye, skin ulcerations, urinary bladder, outer ear canal, nasal and paranasal sinuses, oral cavity, and lower airways via inhalation [16,32]. According to our results, 0.1% NCT showed fungicidal activity with statistical significance and about a 3 log_10_ reduction in viable counts against *C. albicans* isolates within 60 min and *C. non-albicans* within 2 h.

The antimicrobial properties of NCT have been demonstrated by many researchers. A detailed investigation of the activity of 1.0% NCT against *Candida* species was provided by Nagl et al. [20]. According to this work, viable counts of *C. albicans*, *C. krusei*, *C. dubliniensis*, and *C. tropicalis* were reduced significantly by 1–3 log_10_ within 1–2 h at 37 °C in phosphate buffered saline at pH 7.2. *C. glabrata* was found the most resistant species with a decrease of 2 log_10_ after 4 h. Consistent with these results, NCT showed lower fungicidal activity against *C. non-albicans* (especially *C. tropicalis*) than *C. albicans*. Since entrance of the active chlorine of NCT to the cytosol is necessary for the killing of microorganisms [33], it must be assumed that penetration of the substance into *C. glabrata* is prolonged compared to the other *Candida species*.

Of note, in the present study a low concentration of NCT, namely 0.1%, was tested in skim milk and revealed approximately similar killing activity as 1.0% in PBS [18]. This is surprising at first view since chlorine consumption by reducing substances in skim milk will decrease the activity. However, it can be explained by an overcompensation of this consumption by chlorine transfer from NCT to amino compounds in the milk with the formation of corresponding chloramines in equilibrium (transchlorination, transhalogenation) [34,35]. Particularly, the formation of monochloramine (NH_2_Cl) from NCT and ammonium is important, since this compound is more lipophilic and penetrates pathogens better than NCT with clear enhancement of killing activity against bacteria and, particularly, fungi [36,37].

Therefore, transhalogenation significantly increases the microbicidal activity of NCT in the presence of organic matter. Gruber et al. [19] showed that at a concentration of 1% NCT killed bacterial and fungal spores within 10 min and 15 min, respectively, in artificial sputum medium, which mimics the composition of cystic fibrosis mucus. The BA values for fungicidal activity of 1.0% NCT in this study came to 0.71 for *C. albicans* and to 0.37–0.50 for molds. This means that NCT at the clinically preferred concentration of 1% may have similar microbicidal activity as CHX, OCT, and PVP-I in the presence of organic matter occurring in body fluids and exudates. Although the skim milk used in our study was insufficient to imitate body fluids, it has been shown in previous studies that enhancement by body fluids is a specific advantage of NCT since other antiseptics used in human medicine up to date lose activity by organic matter [17]. In acidic milieu, the killing activity of NCT becomes higher, but the effect is more pronounced in bacteria than in fungi [38,39]. Furthermore, one of the main advantages of NCT over other antiseptics is that NCT is a natural antiseptic with exceptional tolerability by the human mucosa [16]. A further important fact for VVC may be that NCT is active against long-term biofilms of up to several months, irrespective of both single and mixed biofilms of bacteria and *C. albicans* [40]. Secretory aspartyl proteinases are one of the major virulence factors of *C. albicans* that have been suggested to play a role in vaginitis [41]. Moreover, secretory aspartyl proteinases of *C. albicans* are downregulated by sublethal concentrations of NCT [20]. These properties of NCT should be considered for adjunctive treatment of *Candida* vaginitis. Although NCT may be used for short-term antisepsis before surgery, extensive studies are needed to evaluate the effect of long-term use of NCT as an antiseptic on the vaginal mucosa, vaginal microbiota, and, especially, vaginal lactobacilli.

Because of the widespread use of fluconazole, resistance to this drug is reported in both *C. albicans* and non-albicans species. According to the results of another study by our group [21] it was found that out of forty-four studied Candida strains, four (two *C. glabrata*, one *C. albicans*) were characterized as susceptible dose-dependent, and two (*C. glabrata*) were found to be resistant to fluconazole. Due to the unspecific oxidizing/chlorinating mechanism of action of active chlorine compounds and confirmed previous studies against different kinds of pathogens, none of the studied Candida strains showed resistance to NCT [37,39,42,43]. Therefore, the use of NCT as an alternative antiseptic is also important in the treatment of VVC caused by fluconazole-resistant Candida strains.

## 5. Conclusions

Since the treatment of VVC is difficult and challenging for both patients and physicians, alternative treatments for these infections, including antiseptics, draw attention. CHX, OCT, PVP-I, and boric acid have been used for years, including for vaginal infections. Based upon all presently available data, including our results, NCT may be a good candidate to be as an endogenous antiseptic with specific advantages and efficacy against *Candida* species isolated from VVC. However, studies on the tolerance of human vaginal epithelium and vaginal microbiota to NCT are needed, especially for long-term use as an antiseptic.

## Figures and Tables

**Figure 1 jof-08-00682-f001:**
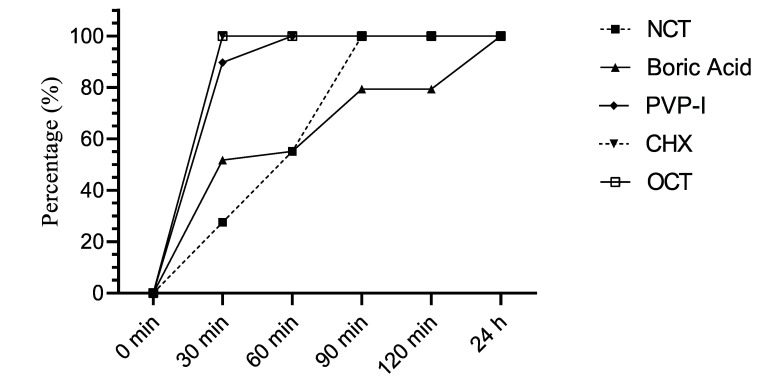
Percentage of *C. albicans* strains (*n* = 29) reduced by ≥4 log_10_ by antiseptics after indicated incubation periods.

**Figure 2 jof-08-00682-f002:**
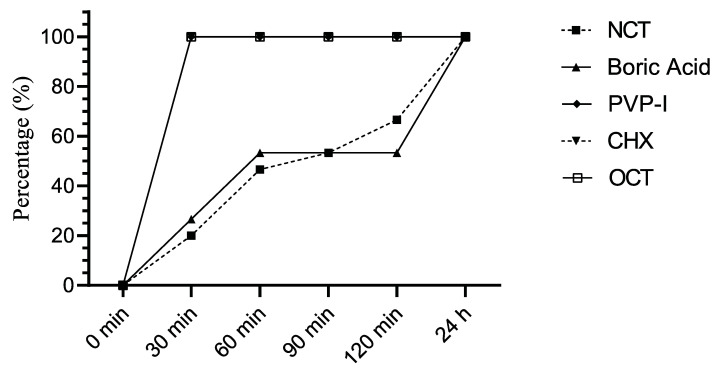
Percentage of *C. non-albicans* strains (*n* = 15) reduced by ≥4 log10 by antiseptics after indicated incubation periods.

**Figure 3 jof-08-00682-f003:**
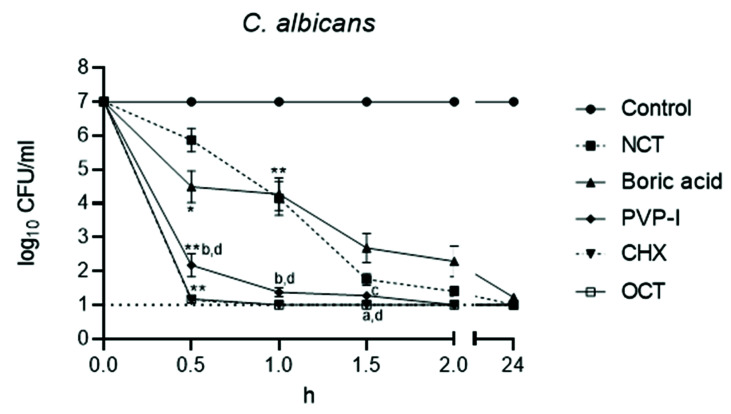
Fungicidal activity of antiseptics against *C. albicans* (*n* = 29) at 37 °C and pH 7. Mean values ± standard error of the mean (SEM); * threshold *p* < 0.05 versus control; ** threshold *p* < 0.01 versus control; ^a^  *p* < 0.05 versus NCT; ^b^  *p* < 0.01 versus NCT; ^c^  *p* < 0.05 versus boric acid; ^d^  *p* < 0.01 versus boric acid; *p* > 0.05 between NCT and boric acid; *p* > 0.05 between PVP-I, CHX, OCT (Kruskal–Wallis test).

**Figure 4 jof-08-00682-f004:**
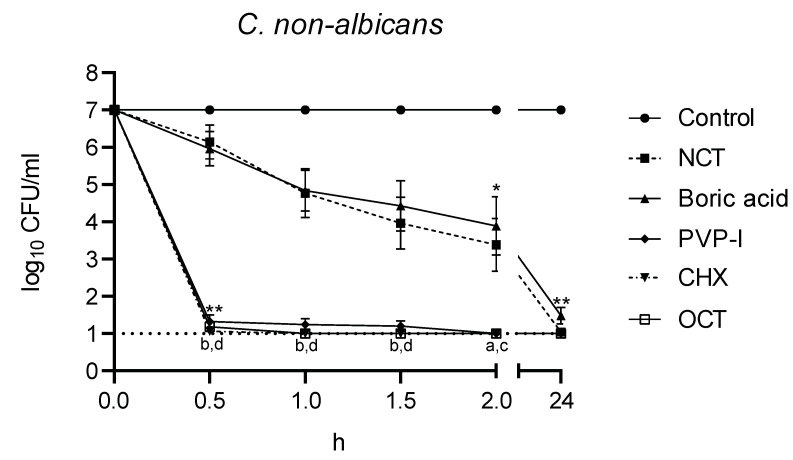
Fungicidal activity of antiseptics against *C. non-albicans* (*n* = 15) at 37 °C and pH 7. Mean values ± standard error of the mean (SEM); * threshold *p* < 0.05 versus control; ** threshold *p* < 0.01 versus control; ^a^ *p* < 0.05 versus NCT; ^b^ *p* < 0.01 versus NCT; ^c^ *p* < 0.05 versus boric acid; ^d^ *p* < 0.01 versus boric acid; *p* > 0.05 between NCT and boric acid; *p* > 0.05 between PVP-I, CHX, OCT (Kruskal–Wallis test).

**Table 1 jof-08-00682-t001:** Total number of test strains and fluconazole resistance profiles.

	Number of Strains	Number of Fluconazole Resistant Strains	Number of Susceptible Dose-Dependent
*C. albicans*	29	-	1 (3.44%)
*C. glabrata*	8	2 (25%)	2 (25%)
*C. krusei*	1	-	1 (100%)
*C. tropicalis*	2	-	-
*C. famata*	1	-	-
*C. kefyr*	2	-	-
*C. lusitaniae*	1	-	-
Total	44	2 (4.54%)	4 (9.09%)

(-): no resistance found.

**Table 2 jof-08-00682-t002:** Fungicidal activity (BA values) of antiseptics.

Antiseptic	*C. albicans* (*n* = 29)	*C. non-albicans* (*n* = 15)
CHX 1.6%	0.1914 ± 0.0014	0.1961 ± 0.0012
OCT 0.08%	0.1890 ± 0.0017	0.1884 ± 0.0017
PVP-I 8%	0.1243 ± 0.0015	0.1593 ± 0.0018
NCT 0.1%	0.0287 ± 0.0001	0.0086 ± 0.0001
Boric acid 8%	0.0132 ± 0.0002	0.0066 ± 0.0001

The higher the BA value, the higher the fungicidal activity; *p* < 0.001 between all single antiseptics for *C. albicans* and *C. non-albicans.*

## Data Availability

Not applicable.

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
