# Peer review of "Comparative Fungicidal Activities of N-Chlorotaurine and Conventional Antiseptics against Candida spp. Isolated from Vulvovaginal Candidiasis"

_jof, 2022, doi:10.3390/jof8070682_

Round 1

Reviewer 1 Report

The manuscript #jof-1738425 "Comparative Fungicidal Activities of N-Chlorotaurine and Conventional Antiseptics Against Candida spp. Isolated from Vulvovaginal Candidiasis" by Hacioglu et al. presentes comprehensive study in line with title. The activity of NCT was compared with conventional antiseptics. Despite the overall good quality of the manuscript I have some issues:

Figure 1 - should be presented as a Table. It would be good to include some background information on the strains - the source, type of infection, patients details.

2.1. CHROM-agar + API test is not enough to distuingiush between Candida spp. Sometimes, strains of some species (C. auris or C. dubliniensis) are incorrectly assigned as C. albicans by those methods. 

2.2. Despite the reference, the very brief, general description of how NCT was obtained should be included. 

In my opinion the presented manuscript should be shortened into a Short Communication instead of an full Article. The study presents preliminary results of the potential activity of the compounds against Candida spp. without a broader background.

Author Response

We are grateful to the reviewer for the valuable comments.

We have taken into account all suggestions and adjusted the manuscript accordingly.

In the following, we provide our point-by-point answers.

The manuscript #jof-1738425 "Comparative Fungicidal Activities of N-Chlorotaurine and Conventional Antiseptics Against Candida spp. Isolated from Vulvovaginal Candidiasis" by Hacioglu et al. presentes comprehensive study in line with title. The activity of NCT was compared with conventional antiseptics. Despite the overall good quality of the manuscript I have some issues:

Figure 1 - should be presented as a Table. It would be good to include some background information on the strains - the source, type of infection, patients details.

Answer: Figure 1 was presented as a Table and strain numbers were written. However, since the number of strains is too much to give as a table, information about the strains is given separately as supplementary material. All strains were collected from patients diagnosed with vulvovaginal candidiasis at the Clinical Microbiology Laboratories of Group Florence Nightingale Hospitals in Turkey. Therefore, only their susceptibilities to various antifungals are given in supplementary material as an additional information.

2.1. CHROM-agar + API test is not enough to distuingiush between Candida spp. Sometimes, strains of some species (C. auris or C. dubliniensis) are incorrectly assigned as C. albicans by those methods.

Answer: The strains in the study were obtained from the Clinical Microbiology Laboratories of Group Florence Nightingale Hospitals. When the strains were obtained by us, they were already identified by the hospital in the Vitek 2 device for diagnosis and treatment. We cultured them onto Chromagar and then reconfirmed with the API Candida. In fact, that’s why we did not think that there was a need to distinguish strains with an extra method, as we got the same results with three different methods.

A text indicating that the same results were obtained with Vitek 2 was added to the method.

2.2. Despite the reference, the very brief, general description of how NCT was obtained should be included.

Answer: A more detailed description has been added in the materials and methods on page 4 line 136-140.

In my opinion the presented manuscript should be shortened into a Short Communication instead of an full Article. The study presents preliminary results of the potential activity of the compounds against Candida spp. without a broader background.

Answer: NCT, is a mild, long-lived oxidant produced by activated human granulocytes and monocytes and thus can be applied to sensitive body parts. The antimicrobial effects is known but studies investigating its efficacy against Candida sp are few. Since every new agent which can support treatments of vulvavaginal candidiasis is invaluable, we think that our study can provide support in this regard. Therefore, the purpose of this study was to determine the in vitro activities of NCT and also the direct comparison of the most important antiseptics used up to date for treatment of this disease. Our study highlights a new and important use of NCT as a vaginal antiseptic. 

Reviewer 2 Report

The manuscript from Hacioglu et al. describes the comparative evaluation of  the antifungal effects of Fungicidal Activities of N-Chlorotaurine and Conventional Antiseptics Against a range of Candida spp. isolates.

The antimicrobial assays were performed against a set of 44 strains.
Despite the relevance of the topic, the anti-Candida action of N-Chlorotaurine is not a new find. In this sense, I would expect that the author would evaluate other aspects related to the action of N-Chlorotaurine: biofilm eradication, evaluation of fungal tolerance, combinatory effects with drugs, insights into the action mechanism. Thieve additional experiments are needed for improve the manuscript. The authors also need to better characterize their isolates by providing the phenotypic profile of resistance for antibiotics. Did you already publish the information about the microbial isolation? Where is the protocol of acceptance in a ethics committee?  The manuscript also have some typos which needs to be corrected (highlighted in the attached pdf file).   Based on these issues, my recommendation is the rejection of this manuscript.

Author Response

We are grateful to the reviewer for the valuable comments.

We have taken into account all suggestions and adjusted the manuscript accordingly.

In the following, we provide our point-by-point answers.

The manuscript from Hacioglu et al. describes the comparative evaluation of  the antifungal effects of Fungicidal Activities of N-Chlorotaurine and Conventional Antiseptics Against a range of Candida spp. isolates.

The antimicrobial assays were performed against a set of 44 strains.

Despite the relevance of the topic, the anti-Candida action of N-Chlorotaurine is not a new find. In this sense, I would expect that the author would evaluate other aspects related to the action of N-Chlorotaurine: biofilm eradication, evaluation of fungal tolerance, combinatory effects with drugs, insights into the action mechanism. Thieve additional experiments are needed for improve the manuscript. The authors also need to better characterize their isolates by providing the phenotypic profile of resistance for antibiotics. Did you already publish the information about the microbial isolation? Where is the protocol of acceptance in a ethics committee?  The manuscript also have some typos which needs to be corrected (highlighted in the attached pdf file). Based on these issues, my recommendation is the rejection of this manuscript.

Answer: The corrections were done which are highlighted in the attached pdf file and the susceptibilities to various antifungals of the strains are given in supplementary material.

In the preliminary studies of the article, sensitivity studies were carried out as well as biofilm studies, but its activity was not found to be sufficient to continue to the study. The most important reason for this is that NCT must be in the presence of an organic substance for antimicrobial activity studies. Therefore, it was decided to conduct antiseptic activity studies. Our study highlights a new and important use of NCT as a vaginal antiseptic.

Reviewer 3 Report

This paper is a continuation of the long series of the articles on NCT produced in Innsbruck by Dr Nagl team. It brings new and complementary in vitro data on NCT and Candida by including vaginal isolates. Discussion and conclusions go too far, as all vaginal antiseptics, maybe with exception of boric acid, destroy vaginal microbiota and especially vaginal lactobacilli, and therefore NCT should not be recommended for long-term treatment of the vaginal infections including recurrent VVC. It should be noted that originally vaginal antiseptics were introduced as short-term vaginal disinfection before surgery. All the tested vaginal antiseptics were reported as negatively active against human vaginal epithelium.  

Number of the tested Candida isolated is rather limited, and the proportion of C.albicans to non-albicans is unfavorable to the latter. It is rather difficult to get valuable data based on 15 strains.          

Minor remarks:

L29: not all vaginal infections are symptomatic and not all symptomatic ones show all three symptoms.

L33: actually, non-albicans VVC accounts up to 10 - 45% of all VVC, thus it is not occasional.

L269: use of skim milk is not sufficiently justified since it contains proteins and saccharides different from those present in vaginal secretion; synthetic cervical mucus would be more appropriate.

L286: skim milk is not a (human) body fluid. It could be used as an organic load in such an in-vitro test but conclusions related to human applications of NCT should be not be drawn on this basis.

L301: again, tolerance of human vaginal epithelium and vaginal microbiota to NCT should be evaluated separately before making such statements.        

Author Response

We are grateful to the reviewer for the valuable comments.

We have taken into account all suggestions and adjusted the manuscript accordingly.

In the following, we provide our point-by-point answers.

This paper is a continuation of the long series of the articles on NCT produced in Innsbruck by Dr Nagl team. It brings new and complementary in vitro data on NCT and Candida by including vaginal isolates. Discussion and conclusions go too far, as all vaginal antiseptics, maybe with exception of boric acid, destroy vaginal microbiota and especially vaginal lactobacilli, and therefore NCT should not be recommended for long-term treatment of the vaginal infections including recurrent VVC. It should be noted that originally vaginal antiseptics were introduced as short-term vaginal disinfection before surgery. All the tested vaginal antiseptics were reported as negatively active against human vaginal epithelium. 

Answer: We are grateful for the reviewer for this point.

The advantage of N-chlorotaurine over other antiseptics is that N-chlorotaurine is a natural antiseptic with exceptional tolerability by the human mucosa. In particular, tolerance studies on vaginal microbiota will definitely be evaluated in further studies. Necessary additions have been made to the discussion page 9, line 327-328 on this subject.

Number of the tested Candida isolated is rather limited, and the proportion of C.albicans to non-albicans is unfavorable to the latter. It is rather difficult to get valuable data based on 15 strains.         

Answer: Since the most cases of VVC, are caused by Candida albicans; followed by non-albicans species, such as C. glabrata, C. tropicalis, C. krusei and C. parapsilosis, we added 29 C. albicans strains and a few non-albicans strains as representative in our study. In any case, we are grateful for the reviewer for this point, which may be a main topic for further studies with NCT.

Minor remarks:

L29: not all vaginal infections are symptomatic and not all symptomatic ones show all three symptoms.

Answer:This sentence was rewritten.

L33: actually, non-albicans VVC accounts up to 10 - 45% of all VVC, thus it is not occasional.

Answer: The sentence was corrected.

L269: use of skim milk is not sufficiently justified since it contains proteins and saccharides different from those present in vaginal secretion; synthetic cervical mucus would be more appropriate.

Answer: The reason for using skim milk is because it meets the requirements of the in vitro antiseptic activity test published by the European Standard. However, it definitely will be considered in future studies, especially in studies that will evaluate the effect of NCT on the vaginal mucosa.

L286: skim milk is not a (human) body fluid. It could be used as an organic load in such an in-vitro test but conclusions related to human applications of NCT should be not be drawn on this basis.

Answer: The situation mentioned in page 9, line 322-323 belongs to a previous study. Limitations about using skim milk in our study were mentioned in discussion.

L301: again, tolerance of human vaginal epithelium and vaginal microbiota to NCT should be evaluated separately before making such statements.   

Answer: Statement was rewritten at line 335-339 and 371-373.

Reviewer 4 Report

The article by Hacioglu et al., describes an interesting compound N-Chlorotaurine (NCT) in treating vaginal candidiasis. Vaginal cavity has a much lower pH than that of the oral environment as a result the vaginal niche presents a more challenging environment for microorganisms to grow. While most common drugs in treating VVC are azoles as described previously, however, evolution to drug resistance to azoles especially in Candida isolated from VVC patients is a rising concern. This calls for characterizing novel antifungals and the present article proposes one such compound with a strong antifungal activity against Candida. 

Here are my comments:

1)In the introduction the authors should highlight that azole drug resistance is a common problem in treating VVC patients. Further, the authors should discuss some known mechanisms of azole resistance that arise while treating VVC patients. The authors can refer to articles like https://pubmed.ncbi.nlm.nih.gov/24962255/ 

and 

https://pubmed.ncbi.nlm.nih.gov/27431223/

This will bolster the importance of the article.

2)As the authors pointed out that vaginal pH is 4.5. Did the authors do the antifungal activity assays at pH 4.5? Please clarify.

3)How were the concentrations of individual antiseptics standardized? Please clarify.

4)Since the authors used several non-albicans species, did any of the non-albicans species showed resistance to NCT?

5)To the reviewer fig 1 should be provided in a table format. Are the MICs to fluconazole known for these strains? If so please mention them in the table. Since NCT or other antiseptics can be used as an alternative to azoles, it is important to know the fluconazole sensitivity of these strains.

Author Response

We are grateful to the reviewer for the valuable comments.

We have taken into account all suggestions and adjusted the manuscript accordingly.

In the following, we provide our point-by-point answers.

The article by Hacioglu et al., describes an interesting compound N-Chlorotaurine (NCT) in treating vaginal candidiasis. Vaginal cavity has a much lower pH than that of the oral environment as a result the vaginal niche presents a more challenging environment for microorganisms to grow. While most common drugs in treating VVC are azoles as described previously, however, evolution to drug resistance to azoles especially in Candida isolated from VVC patients is a rising concern. This calls for characterizing novel antifungals and the present article proposes one such compound with a strong antifungal activity against Candida.

Here are my comments:

1)In the introduction the authors should highlight that azole drug resistance is a common problem in treating VVC patients. Further, the authors should discuss some known mechanisms of azole resistance that arise while treating VVC patients. The authors can refer to articles like https://pubmed.ncbi.nlm.nih.gov/24962255/

and https://pubmed.ncbi.nlm.nih.gov/27431223/

This will bolster the importance of the article.

Answer: Azole drug resistance was highlighted in the introduction according the manuscripts that reviewer suggested.

2)As the authors pointed out that vaginal pH is 4.5. Did the authors do the antifungal activity assays at pH 4.5? Please clarify.

Answer: These tests were done at the standard pH of 7 since the main aim of the study was to test a larger number of different Candida species and not to test different conditions. It is right that the pH has an influence on the microbicidal activity of, for instance, NCT. In acidic milieu, the killing activity of NCT becomes higher, but the effect is more pronounced in bacteria than in fungi [1-2], so that such tests would not have influenced the main outcome of the that all the isolates are susceptible. In any case, we are grateful for the reviewer for this hint, which may be a main topic for further studies with several antiseptics.

An accoding hint has been added in the discussion on page 9 line 325-326

3)How were the concentrations of individual antiseptics standardized? Please clarify.

Answer: Standard concentrations of antiseptics that are within the range of use or possible use concentrations were chosen, this information was added at page 4 line 141-142. For NCT, the chosen concentration of 0.1% is relatively low and demonstrated activity against all strains. Mind that MICs and MBCs do not make much sence for antiseptics (except for very specific scientific questions) since the general application concentration is markedly higher. 

4)Since the authors used several non-albicans species, did any of the non-albicans species showed resistance to NCT?

Answer: No. This is logical and due to the unspecific oxidizing / chlorinating mechanism of action of active chlorine compounds and confirms previous studies against different kinds of pathogens, e.g. [2-5].

We have added this in the discussion on page 10 line 350-354.

5)To the reviewer fig 1 should be provided in a table format. Are the MICs to fluconazole known for these strains? If so please mention them in the table. Since NCT or other antiseptics can be used as an alternative to azoles, it is important to know the fluconazole sensitivity of these strains.

Answer: Figure 1 was presented as a table and MICs of fluconazole was added in it. MICs of all studied antifungals which was done another study by our group [6], were given as supplementary material. An accoding point has been added in the discussion on page 10 line 346-350.

  1. Nagl M and Gottardi W. In vitro experiments on the bactericidal action of N-chlorotaurine. Hyg. Med. 1992; 17: 431-439.
  2. Nagl M, Lass-Flörl C, Neher A, Gunkel AR and Gottardi W. Enhanced fungicidal activity of N-chlorotaurine in nasal secretion. J. Antimicrob. Chemother. 2001; 47: 871-874.
  3. Anich C, Orth-Höller D, Lackner M and Nagl M. Microbicidal activity of N-chlorotaurine against multiresistant nosocomial bacteria. J. Appl. Microbiol. 2021; DOI: 10.1111/jam.15052. DOI: 10.1111/jam.15052.
  4. Lackner M, Binder U, Reindl M, Gönül B, Fankhauser H, Mair C and Nagl M. N-chlorotaurine exhibits fungicidal activity against therapy-refractory Scedosporium species and Lomentospora prolificans. Antimicrob. Agents Chemother. 2015; 59: 6454-6462. 10.1128/AAC.00957-15.
  5. Lackner M, Rössler A, Volland A, Stadtmüller M, Müllauer B, Banki Z, Ströhle J, Luttick A, Fenner J, Stoiber H, von Laer D, Wolff T, Schwarz C and Nagl M. N-chlorotaurine, a novel inhaled virucidal antiseptic is highly active against respiratory viruses including SARS-CoV-2 (COVID-19) in vitro. Emerging Microbes and Infection. 2022; 11: 1293-1307. 10.1080/22221751.2022.2065932.
  6. Hacioglu M, Guzel CB, Savage PB, Tan ASB. Antifungal susceptibilities, in vitro production of virulence factors and activities of ceragenins against Candida spp. isolated from vulvovaginal candidiasis. Med Mycol. 2019;57(3):291-299. doi:10.1093/mmy/myy023

Round 2

Reviewer 1 Report

The manuscript was revised according to most of the suggestions. However, I still believe the manuscript should be shortened into a Short Communication instead of an full Article.

Reviewer 2 Report

Although the authors have made the adjustments suggested in the text in the first reviewer report, they did not provide any new information obtained from new assays. The manuscript still has few innovative features to be considered for publication in the high standard journal as Molecules. Made they should try to submit the manuscript as a short communication.   As indicated in the first reviewer report, the antifungical or antiseptic effects of N-chlorotaurine is not a new topic. Indeed, some reviewers have been published in this subject:   GOTTARDI, Waldemar; NAGL, Markus. N-chlorotaurine, a natural antiseptic with outstanding tolerability. Journal of antimicrobial chemotherapy, v. 65, n. 3, p. 399-409, 2010.  
NAGL, Markus; ARNITZ, Roland; LACKNER, Michaela. N-Chlorotaurine, a promising future candidate for topical therapy of fungal infections. Mycopathologia, v. 183, n. 1, p. 161-170, 2018.